# Being a Parent of Children with Disabilities during the COVID-19 Pandemic: Multi-Method Study of Health, Social Life, and Occupational Situation

**DOI:** 10.3390/ijerph20043110

**Published:** 2023-02-10

**Authors:** Noémie Fortin-Bédard, Naomie-Jade Ladry, François Routhier, Josiane Lettre, David Bouchard, Béatrice Ouellet, Marie Grandisson, Krista L. Best, Ève-Line Bussières, Marie Baron, Annie LeBlanc, Marie-Eve Lamontagne

**Affiliations:** 1Center for Interdisciplinary Research in Rehabilitation and Social Integration, Centre Intégré Universitaire de Santé et de Services Sociaux de la Capitale-Nationale, Quebec, QC G1C 3S2, Canada; 2Department of Rehabilitation, Université Laval, Quebec, QC G1V 0A6, Canada; 3Department of Psychology, Université du Québec à Trois-Rivières, Trois-Rivieres, QC G8Z 4M3, Canada; 4VITAM Research Center on Sustainable Health, Centre Intégré Universitaire de Santé et de Services Sociaux de la Capitale-Nationale, Quebec, QC G1J 2G1, Canada; 5Department of Family and Emergency Medicine, Université Laval, Quebec, QC G1V 0A6, Canada

**Keywords:** COVID-19, people with disabilities, rehabilitation, qualitative

## Abstract

Parents of children with disabilities face challenges in their daily lives, but little is known about their experience of the COVID-19 pandemic. The objective of the study was to explore the experiences of parents of children with disabilities during the COVID-19 pandemic in Quebec, Canada. Forty parents of children with disabilities from Quebec, Canada (mean [SD] age: 41.2 [6.7]; 93% women) were selected from the Ma Vie et la pandémie (MAVIPAN) study. All 40 parents completed the MAVIPAN online questionnaires including the Depression, Anxiety and Stress Scale (DASS-21), Warwick–Edinburgh Mental Wellbeing short 7-item scale (WEMWBS), Social Provisions Scale-10 item (SPS-10), and the UCLA Loneliness Scale (UCLA-LS). A multi-method analysis was used to summarize questionnaires and thematically explore parents’ experiences. Parents reported deterioration in their mental (50.0%) and physical (27.5%) health, with moderate levels of depression, stress, and anxiety, yet moderately positive well-being. Additional experiences included reduction in available supports (71.4%) and feelings of social isolation (51.4%). Our results highlighted reduced mental and physical health, limited and modified access to certain services, and reduction of social supports for some parents of children with disabilities. Health professionals, policymakers, and governments should be mindful of these challenges experienced by parents of children with disabilities.

## 1. Introduction

In 2018, over 600,000 Canadian parents provided care to their child with long-term health conditions caused by a disability [1]. Like any other child, children with disabilities require care, but given the complexity of their needs, parenting a child with disability is often more physically and psychologically demanding [2,3,4]. In fact, it involves many extra responsibilities, such as learning to manage the condition and its associated symptoms, attending healthcare and rehabilitation appointments, providing assistance in everyday life activities, adapting the environment and the family routine, or advocating for children to get the support and opportunities they need to function and participate in the community [2,3,4]. The significant burden associated with these constant efforts is a source of anxiety [5,6], depression [7], and time out of work, which may lead to limited income [8]. Resources and services are needed to better support parents of children with disabilities [9] and reduce burden.

According to Bronfenbrenner’s ecological systems theory [10], several resources and services in the environment can support parents of children with disabilities such as help from friends and family (microsystem) [11], healthcare and respite services (exosystem) [12], and the Government of Canada Child Disability Benefit (macrosystem) [13]. However, many parents of children with disabilities experience difficulties accessing or using these resources to meet the needs of their children [1,11,14]. Resources access may be further limited by various individual, social, physical, organization and systemic barriers [5]. For example, such barriers may be financial (e.g., private rehabilitation services to compensate limited availability of public service) or organizational due to the bureaucracy (e.g., system-centred rather than a system based on the child needs) [5,15]. Parents express frustrations regarding their difficulties to access resources, a lack of support, and an increased burden [14].

The COVID-19 pandemic may have made access to these resources more complex and disrupted the daily lives of families of children with disabilities. Indeed, protection and isolation measures had brought significant changes that had directly affected the daily lives of families with children with disabilities such as working from home, remote education, the cessation of children’s activities in the community, and the reorganization of the care and services offered [16]. There have been reports of reduced mental well-being among parents, which increased levels of anxiety and fear [17,18]. For example, Cacioppo et al. revealed that parents of children with disabilities identified that mental load was the main daily difficulty during the COVID-19 lockdown [19]. Furthermore, parents revealed that they were not satisfied with the services their child received during the pandemic, such as telehealth services for therapy [17]. Despite emergent evidence of pandemic disruptions on families with children with disabilities, few studies have addressed specific impacts of the pandemic and the isolation and protective measure on the parents of children with disabilities. Therefore, the general objective of this study was to investigate the experiences of parents of children with disabilities during the COVID-19 pandemic between March 2020 and July 2021 in Quebec, Canada. More specifically, we aimed to (1) document the physical health and mental wellbeing of parents of children with disabilities, their social life, and their occupational situation during the COVID-19 pandemic; (2) qualitatively explore the experience of the parents of children with disabilities.

## 2. Materials and Methods

### 2.1. Design and Participants

The sample for this exploratory, cross-sectional multi-methods (independent concurrent quantitative and qualitative data collection) study was drawn from the Ma vie et la pandémie (MAVIPAN) study, a longitudinal prospective cohort study documenting how individuals, families, healthcare workers, and healthcare organizations are affected by the pandemic and how they adapt (ClinicalTrials.gov Identifier: NCT04575571) [20]. Participants indicated in the consent form whether they accepted to be recontacted for qualitative interviews. Complete details of the MAVIPAN study protocol, have been published [20]. MAVIPAN recruitment began on 29 April 2020, through the study website (www.mavipan.ca), mainstream media, social media, and mass diffusion across healthcare centers and universities. MAVIPAN participants were (1) at least 14 years of age, (2) understood French or English, and (3) lived in Quebec, Canada [20]. To be eligible to our study, participants should additionally have indicated that they were a caregiver of children with special health needs under the age of 18 on the MAVIPAN online questionnaire. As of July 2021, 3155 participants were recruited in the MAVIPAN study, including 40 parents of children with disabilities. Participants in this subset who accepted to be recontacted for qualitative interviews were reached by email to complete an interview aiming to document their experiences associated to the COVID-19 pandemic situation. The MAVIPAN study was approved by the Research Ethics Board in population health at the Centre intégré universitaire de santé et de services sociaux de la Capitale-Nationale (#2021–2043).

### 2.2. Quantitative Data: Online Questionnaire

Sociodemographic data of the participants, including health and employment situation based on Canadian Community Health Survey, were collected from the MAVIPAN online questionnaires [21]. Participants completed questionnaires between 30 April 2020 and 4 February 2021. This period corresponded to different protective measures, depending on the administrative region of the participant’s residence. In Quebec, a curfew was put in place from 9 January to 8 February 2021, except in two regions [16]. Thereafter, a gradual reopening was put in place until May 2021, depending on the region of residence. A subsample of participants completed an additional questionnaire regarding their role as caregivers during the pandemic that was added during the second COVID-19 wave. The emotional states of depression, anxiety, and stress was assessed using the Depression, Anxiety and Stress Scale (DASS-21) [22]. The Cronbach’s alphas for the English version of the DASS-21 are 0.94, 0.87, and 0.91 for the Depression, Anxiety, and Stress subscales, respectively [23]. The French version of the DASS-21 was used in the present study [24]. The scale is divided into three subscales comprising seven questions about depression, anxiety, and stress; each subscale has a maximum score of 42 points and the total score is 126 points [22]. Mental well-being was assessed using the French validated version of the Warwick–Edinburgh Mental Wellbeing short 7-item scale (WEMWBS; Cronbach’s alpha = 0.91 [25]. Scores range from 7 to 35 with higher scores indicative of positive mental wellbeing [25]. The French validated Social Provisions Scale-10 item (SPS-10; SPS-10; Cronbach’s alpha = 0.88) was used to access relationships with friends, family members, co-workers, and community members [26]. The sum of the scores ranges from 10 to 40 with higher scores indicative of higher levels of social support. Four items of the French version of the UCLA Loneliness Scale (UCLA-LS; Cronbach’s alpha = 0.87) were used to assess the level of solitude of the participants [27].

### 2.3. Qualitative Data: Interviews

A semi-structured interview guide, including 8 open-ended questions, was developed by a multidisciplinary team of researchers to thoroughly explore the quantitative findings related to the experiences of the parents of children with disabilities. Questions were related to the impacts of the protective measures on the family, their child with a disability, their daily routine, social and community support, on their leisure activities, the adaptation strategies they developed, and their perception of the future. The interview guide was not pilot tested. The complete interview guide is available as Appendix A.

Interviews were conducted between 29 December 2020 and 8 April 2021 by telephone or videoconference. Interviews were audio or video recorded and transcribed verbatim by a research assistant. Interview duration ranged from 15 to 55 min.

### 2.4. Data Analysis

Sociodemographic data were summarized using descriptive statistics (mean, standard deviation, frequency, percentage). The mean (SD) of DASS-21 and WEMWBS were summarized. Item-specific frequencies were calculated for the SPS-10 and UCLA-LS. All quantitative data were analyzed using the Statistical Package for the Social Sciences (SPSS; IBM version 27). Qualitative data were coded using a mixed inductive and deductive approach [28]. Braun and Clark thematic analysis [29] was completed by three authors (NJL, NFB, DB) using N’Vivo software (Release 1.0). Some codes were defined prior to analysis according to the topic surveyed in the interview guide and other codes were added, modified, and deleted according to emerging theme of the interviews. Three interviews were initially coded by one author (NJL) and then about 10% of the coding was verified (NFB, DB) to ensure agreement. A research associate (DB) supervised the coding and the three data coders (NJL, NFB, DB) met regularly to discuss the interpretation of the current data and to generate themes. The analysis was conducted in the original language to limit interpretation of translations.

The results were organized according to the environmental levels of the Bronfenbrenner’s theory of ecological systems [10], which facilitated consideration of the personal characteristics of the parents, the impacts of the pandemic on their immediate systems (such as the role of parents of children with disabilities, employment situation, and social life) and the interrelation between these systems, and the parents’ characteristics to focus on meaningful dimensions of the experience of the parents of children with disabilities during the COVID-19 pandemic [10]. The quantitative and qualitative results are presented concomitantly to facilitate deeper exploration of the findings.

## 3. Results

Forty parents of children under the age of 18 with disabilities from Quebec (QC, Canada) participated in the present study. Among them, a subsample of 16 completed an additional questionnaire specific to their role as a caregiver and five were interviewed.

The mean (SD) age of the 40 parents was 41.2 (6.7) years and they were mostly mothers (92.5%) with a university diploma (60%). The parents had 1.9 children on average who were a mean (SD) of 10.4 (4.0) years of age. The 16 parents who completed the additional questionnaire were 42.2 (6.5) years of age and were mostly mothers (87.5%) with a university diploma (62.6%). These parents had 1.7 children on average and the mean (SD) age of these children was 12.1 (3.3) years old. According to the parents, children had a developmental (43.8%), mobility (12.5%), learning (12.5%), intellectual (12.5%), mental health (6.3%), or other (12.5%) disability. Table 1 presents sociodemographic characteristics (*n* = 40) parents of children with disabilities and the subsample (*n* = 16) parents who completed the questionnaire specific to their role as caregivers. Detailed characteristics of the subsample (*n* = 5) of mothers who were interviewed are presented in Appendix A.

Four main themes presenting the impacts of the pandemic on parents of children with disabilities were identified, including the following: (1) physical health and mental well-being; (2) role as parents’ caregivers; (3) employment situation; and (4) social life.

### 3.1. Physical Health and Mental Well-Being

Parents experienced moderate levels of depression, anxiety, and stress according to the DASS-21 (Table 2).

Eleven participants (27.5%) reported that their physical health was a little worse or much worse now compared with before the COVID-19 pandemic. Of note, nine participants (21.4%) declared having significant and persistent impairments or disabilities that may present barriers to performing their daily activities (e.g., asthma, chronic pain). During the interview, a parent attending university reported that homeschooling reduced the amount of time to relax, which increased fatigue:

*[...] Often what I observed, I could leave the University with a problem, a small walk of 30–45 min [to go back to] home, go to eat, go to relax, then I reread here and there I have like another point of view, and it goes better. But now I’m not able to do that anymore, making us more tired. We are globally more tired, more exhausted* (participant 3; a mother of two children working in the health and social sector and attending school).

In addition, 20 participants (50.0%) reported that their mental health was a little worse or much worse now compared with before the COVID-19 pandemic. Of note, 15 (37.5%) parents self-identified as having a mental health challenge (e.g., mood disorder, anxiety disorder, or sleep disturbance). During the interview, a mother mentioned having deteriorated mental health during the first wave:

*It’s difficult for the parents, my mental health in the first wave was really not good, but now it is good* (participant 2; a mother of two children working full-time in the health and social sector).

In addition, a mother with a child who was vulnerable to significant complications from COVID-19 reported the added stress of not transmitting the virus to her child:

*Last March, in fact, there was quite a bit of stress because my child has health problems other than ADHD, which meant that two of his specialists said that he should not catch the covid, because he has an autoimmune and pulmonary disease. So, it was a great stress to do everything possible to prevent him from getting it. So, it was a bit of a freak day for him not to get it* (participant 1; a mother of two children working full-time in the health and social sector).

Regarding mental well-being, the mean (SD) score of the WEMWBS was 26.3 (3.9) over 35 corresponding to a fairly positive mental wellbeing. Parents were asked to project how they think they may feel about their life in general one year from the time the questionnaire was completed on a scale of 1 to 10 (1 = very dissatisfied; 10 = very satisfied). Most of the parents (92.2%) reported they would be at 5 or more on the scale in one year. While most parents would be at least somewhat satisfied with their life, a participant mentioned that normal life would not return as it was before the pandemic, but that they would adapt on several levels:

*In fact, I imagine that normal life will not return as it was, like in 2019. However, we will have adapted on several levels, we will eventually get a social life again […]. The kids are going to end up having a party and having friends back, and then I think it’s okay, you just have to be a little patient* (participant 4; a mother of three children living as a single parent and in a parent leave).

Similarly, another participant mentioned that, in the long term, it will be wonderful, it will be great and that they can’t wait for it (i.e., the pandemic) to be over, but that they expect the next few weeks will be challenging.

### 3.2. Role as Parents’ Caregivers

Sixteen parents responded to the questions specific to the effect of the pandemic on their role as parents of children with disabilities (Table 3).

Most parents (71.4%) disagreed or strongly disagreed with the affirmation that the changes in the organization and the provision of services did not have an impact on the support they received in response to their request. For example, a participant expressed that provision of services was very difficult during the pandemic because they could not have access to the normal care and there were often added delays. They were already confronted with the cumbersomeness of accessing healthcare, but the pandemic increased challenges. In particular, heightened challenges increased the mental load that parents already perceived related to becoming an expert in their child’s disability:

*That was very difficult, because we could not have the normal care, and then delays were added too. Already, as we see a lot of doctors, well we are also confronted with the cumbersome of the process, there it added a layer with the COVID-19. […] As a parent it is one thing, but as a parent of a child with disabilities who has always been confronted with these questions, it adds to the small layer of mental load that parents must have which manages everything. Parents who must become a bit of an expert in children’s disease, especially in the context of a rare disease* (participant 5; a mother of two children working full-time).

In addition, most of the participants (80%) agreed or strongly agreed that the suspension of activities among people with disabilities had a big impact on many families needing support. Furthermore, 46.7% of the parents indicated that the suspension of weekend respite services had increased the burden of caring for their child, for them and for other family members.

Remarkably, almost all participants (93.3%) reported that they were able to fulfill their child’s needs. For instance, a participant mentioned having set up a more flexible schedule with expectations adapted to the needs of the child:

*I made [my child] shorter days because I still believe that children like mine are unrealistic to put them through 8 to 4 social interaction days. They have more fragile periods, so we should make them short days with more realistic challenges for them. It allowed me to adjust to all these mood swings* (participant 2; a mother of two children working full-time in the health and social sector).

Finally, 46.7% of the participants disagreed or strongly disagreed that health and social services workers were able to respond to the needs of their child despite the changes in the organization of services provided. In this regard, during the first wave, a participant would have liked more involvement from the school community, especially while the school was at home:

*The weekdays were very disjointed, a lot more [use] of tablets, the school didn’t take much on* (participant 1; a mother of two children working full-time in the health and social sector).

### 3.3. Employment Situation

Since the beginning of the COVID-19 pandemic, almost half of the participants (45.5%) reported they worked from home. In this regard, a participant expressed that their daily life was like many parents, where they tried to survive with the work that continued and the schools that were shut down:

*Initially, of course, the schools were closed, so our daily life was like many where we tried to survive with the work that continued and the school that was shut down. We had split the day in two, my partner and I, to try to help the children as much as possible with school. They had teachers who were involved anyway. Both go to school in normal classes, but with different resources that are more or less adequate. The children had maybe an hour of class per day, for which they were more or less autonomous. An 8-year-old and a 10-year-old, with [disabilities], we couldn’t just leave them in front of the computer. It was often with the laptop, we try to work as much as we can with the child next to it, and the rest of the day was spent doing activities* (participant 5; a mother of two children working full-time).

Indeed, participant could not simply leave the children in front of the computer, instead they tried to work as closely to the child as they could:

*It is as if the health network has done me a little extra to allow me to work at home when it was not so allowed and my partner too. So, it’s like the two of us had an extra being able to work at home but both of us had 40 h weeks to do with a child who needs constant supervision plus another 8 years old. [My child] has a very very very hyperactive profile. Which meant that we were both at home and it was extremely difficult* (participant 2; a mother of two children working full-time in the health and social sector).

In addition, the participant reported that they suddenly lost their support network and had to establish a structured schedule for their child. While this participant’s superiors were flexible and allowed her to work from home, her workload was not reduced. Additionally, the privilege of working from home had to be hidden from other coworkers, which made the participant uncomfortable. Over half of the participants (54.6%) agreed or strongly agreed that they could lose their main job or their main source of self-employment income over the next four weeks. In this respect, 53.2% of the parents were a little, very, or extremely worried about the impact of the pandemic on their personal finances. For instance, their ability to meet their basic needs, such as rent, mortgage payments, utilities, and groceries.

### 3.4. Social Life

Regarding the level of solitude of the parents during the pandemic, 62.8% of the participants never or rarely felt a lack of companionship, and 60% of the participants never or rarely felt left out (Table 4).

In fact, one participant reported doing more family activities, playing more board games and having more opportunities to be with family, which cultivated stronger bonds. However, 51.4% of the participants felt isolated from social networks outside the home. For instance, a participant reported that people no longer called because her social network had gradually crumbled during the pandemic. Last summer when the protective measures were reduced, the participant expressed that she again felt forgotten. For this reason, the parent had apprehension regarding the relaxing of lockdown and protective measures.

In addition, a participant mentioned suffering less from social isolation, since everyone else was also isolated. Accordingly, the participant felt like everyone else which felt good:

*I have suffered tremendously from social isolation because you haven’t been able to socialize with my boy that many years for years. It was really hard for me, and then it’s like I’m used to it, I’ve learned to be socially isolated, because it’s too difficult to socialize with it. [...] Right now I suffer almost less from social isolation because I feel like everyone else because I used to feel unequal never to go out. But now, I mean, I’m like everyone else so it looks like it’s almost better* (Participant 2; a mother of two children working full-time in the health and social sector).

Some parents (86.6%) reported that the use of communication technologies and online social network enabled them to maintain relations with family and friends and other members of their circle. In addition, a videoconferencing platform was also used by one participant so that the participant’s mother could supervise the child’s school while working. One participant mentioned being surrounded by family and never feeling alone, which was desirable for this participant. In contrast, a participant reported a less contact with her children:

*There was no real planned activity, it was in the morning you get up, you go to the computer, you do your classes and after that it’s like there’s a… you don’t see them the rest of the day either, they don’t go out, everyone stays in their own little room. It takes away the social contact outside, but even inside everybody stays in their own little room and you don’t see each other much* (participant 3; a mother of two children working in the health and social sector and attending school).

Regarding their social life during the pandemic (Table 5), most parents (87.5%) agreed or strongly agreed that there were people they could count on in an emergency. Nevertheless, 17.1% of the participants reported disagreeing or strongly disagreeing with the affirmation that there are people they can depend on to help them if they really need it. In fact, a participant reported that their main source of support is usually the immediate family (i.e., grandparents and friends). In the context of the pandemic, the immediate family could not provide this support, because they had to avoid exposing the children to the virus.

Finally, according to another participant, the government’s decisions to allow natural caregivers to get involved with people with disabilities at some point during the pandemic allowed to get help and to be organized.

A summary of pandemic impacts on parents life according to Bronfenbrenner’s ecological systems theory [10] is presented in Figure 1.

## 4. Discussion

The general objective of this study was to investigate the experiences of parents of children with disabilities during the COVID-19 pandemic between March 2020 and July 2021 in Quebec, Canada. More specifically, we aimed to (1) document the physical health and mental wellbeing of parents of children with disabilities, their social life, and their occupational situation during the COVID-19 pandemic; and (2) qualitatively explore the experience of the parents of children with disabilities. The policy regarding the protection and isolation measures during the pandemic (i.e., exosystem) had impacts on the parents (microsystem), such as the physical health and mental well-being and their immediate system (e.g., their role as caregivers of children with disabilities, their social life, and their job). Our findings suggested that the interrelation between the impacts on these systems could considerably increase the difficulties experienced during the pandemic by these parents.

Studies conducted before the pandemic reported that parents of children with disabilities have higher risk of depression or another mental health problem compared to parents of children without a disability [30,31]. In the present study, half of the parents of children with disabilities reported their mental health worsened compared with before the pandemic. This result is particularly concerning considering that at the time of the data collection, 37.5% of parents also self-identified as having a mental health problem, suggesting that additional focus may be critical for this population in the context of the pandemic. In this regard, our findings are also consistent with other studies conducted during the COVID-19 pandemic [17,19]. For example, one study highlighted that parents of children with disabilities reported higher mental load and feelings of helplessness during the pandemic [19]. Similarly, Masi et al. reported that the COVID-19 pandemic impacted the well-being of some parents of children with disabilities [17]. In this study, we explored the burden experienced during the pandemic by parent caregivers of children with disabilities. Family caregivers could also care for older individuals with special health needs, and their experience could be different than the experiences of parents of children with disabilities in the present study. Nevertheless, a rapid systemic review reported that family caregivers of individuals with Alzheimer’s disease and other dementias reported a high increase level of burden, anxiety, depression, and distress [32], which is consistent with our findings.

The protection and isolation measures mandated during the pandemic have modified or limited access to certain services for children with disabilities and their parents. In the present study, most parents reported that changes in the organization and the provision of services impacted their supports requested and received. These observations are consistent with previous findings, which suggested that some services (e.g., rehabilitation services for children with disabilities) were massively interrupted, posing a major concern for parents and people with disabilities [19,33]. According to Strunk et al., the pandemic exacerbated pre-existing barriers to accessing quality health for children with disabilities [34]. Moreover, parents indicated significant change in supports or services available [35], and they were not satisfied with services received during the pandemic [17]. A study highlighted the importance of considering the preferences of families of children with disabilities in providing telehealth services tailored to the unique needs of children with disabilities and their families [36]. Even before the pandemic, nearly half of these parents did not receive all of the social or financial support requested and almost half reported that the suspension of weekend respite services had increased the burden of caring for their child, for themselves, and for other family members [1]. As evidenced by a systematic review, respite care is critical for reducing the familial stress and improving parent–child relationships [37]. Therefore, the pandemic has shed light on the need for innovative strategies and services to improve priority access to services for children with disabilities and their parents. This also highlighted the need to offer respite services for parents during the pandemic and in the future.

Most participants (68%) were employed as essential servants during the pandemic, including 33% working in health and social services. Of note, the list of essential government services in Quebec includes priority health and social services and public security services [38]. Several studies have reported that healthcare workers perceived heightened stress, anxiety, and depression during the pandemic [39,40]. Moreover, healthcare workers expressed concern about transmitting COVID-19 to family [39,41], which may partially explain our findings.

It is important to note that more than half of the children had developmental and intellectual disabilities, as Turk et al. found that developmental and intellectual disabilities were associated with poorer COVID-19 outcomes, especially at younger ages [42]. In this context, fear of transmitting COVID-19 to loved ones may increase for parents of children with disabilities working in the health and social services sector. Furthermore, given that almost half of the parents (44%) were working from home during the pandemic, and because schools were closed due to the pandemic, parents had to homeschool their children. In Canada, this situation increased daily responsibilities and parental tasks, yet parents had to continue to work [43]. Despite an increase in responsibility to their children, parents in this study indicated that their workload was not decreased. In this regard, a study reported that school closures had an impact on the physical and mental well-being of parents of children with disabilities due to the pressure of homeschooling and the pressure of dealing with too many jobs [44]. The working situation and homeschooling of children with disabilities thus increased the burden of parents and may have influenced their individual condition and coping abilities. Consequently, future studies should identify relevant measures and explore their impact to better support parents and reduce the mental load during homeschooling.

Finally, the degree of social isolation perceived during the pandemic by parents of children with disabilities was variable. More than half (51.4%) of the parents felt isolated from others during the pandemic. To the best of our knowledge, studies exploring the perceived degree of social isolation of parents of children with disabilities are scarce. Consistent with our findings, a study conducted in the United Kingdom with parents of children without disabilities highlighted that 58% of these parents reported feeling isolated from others during the pandemic [45].

The limitations of this study must be considered when interpreting the results. First, a small convenience sample completed the qualitative interviews, which was not representative of all parents of children with disabilities in Quebec (e.g., a high proportion were essential workers). Additionally, only five participants completed interviews which limited data saturation. However, the multi-method design allowed for an in-depth exploration of perceived challenges and social participation experiences beyond quantitative assessment. Second, the sample was primarily composed of mothers and, accordingly, does not accurately reflect the experience of fathers. Of note, fathers are often underrepresented in research involving children [46]. Third, questionnaires and interviews were completed over many months (i.e., April 2020 to February 2021 for questionnaires, December 2020 to April 2021 for interviews), which may have influenced the perceived experiences of parents of children with disabilities throughout the pandemic. In addition, missing data for quantitative questionnaire may have caused bias. Our small sample included parents of children with several types of disabilities, thus we could not differentiate how disability type may have impacted parents’ experiences during the pandemic. Finally, healthcare is regulated provincially and there is great variability in healthcare system and protection and isolation measures related to COVID-19 throughout Canada, which may limit transferability of our findings.

## 5. Conclusions

This study contributes to the body of knowledge. Our results suggested that the pandemic impacted the mental health of some parents of children with disabilities and limited access to certain services. These difficulties, combined with having essential service jobs in healthcare, feeling isolated from others, and homeschooling have presented critical challenges related to being a parent of children with disabilities during the pandemic that should be addressed. Our results provide an opportunity to suggest changes to better support parents of children with disabilities during and after the pandemic. Health professionals, policymakers, and governments should be mindful of the challenges of parents of children with disabilities highlighted in the present study.

## Figures and Tables

**Figure 1 ijerph-20-03110-f001:**
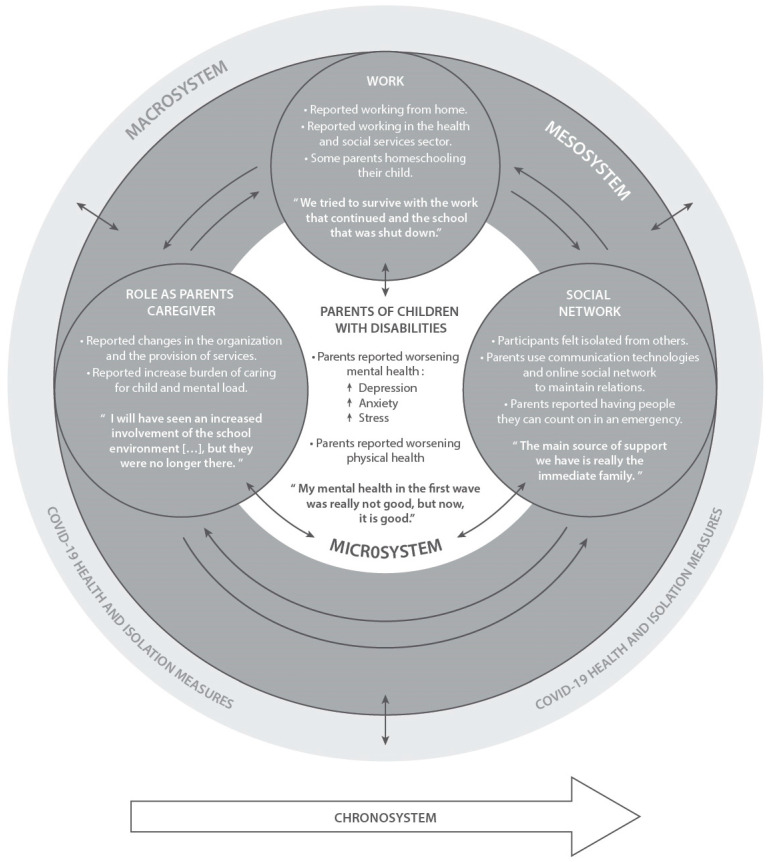
Summary of pandemic impacts on parents’ life according to Bronfenbrenner’s ecological systems theory.

**Table 1 ijerph-20-03110-t001:** Characteristics of the parents of children with disabilities from Quebec province (Canada) during the COVID-19 pandemic *.

Characteristics	All Participants (*n* = 40)	Parents Who Completed a Questionnaire Specific to Their Role as Caregivers (*n* = 16)
Age, years (mean [SD])	41.2 [6.7]	42.2 [6.5]
Sex		
Women	37 (92.5%)	14 (87.5%)
Men	3 (7.5%)	2 (12.5%)
Living with children under 18 years old	38 (95.0%)	16 (100%)
Number of children (mean [SD])	1.9 [2.0]	1.7 [2.0]
Age of children, years (mean [SD])	10.4 [4.0]	12.1 [3.3]
Family situation		
I live with my child/children from my current partner	30 (75.0%)	11 (68.8%)
I live with my child/children from a former partner	9 (22.5%)	5 (31.3%)
I live with a stepchild/stepchild of my partner from a former union	1 (2.5%)	1 (6.3%)
Living as a single parent	6 (15.0%)	3 (18.8%)
Highest diploma, trade certificate or degree completed		
High school	6 (15.0%)	2 (12.5%)
College	10 (25.0%)	4 (25.0%)
Undergraduate university degree	14 (35.0%)	7 (43.8%)
Graduate university degree	10 (25.0%)	3 (18.8%)
Attending school, college, or university		
No	30 (75%)	10 (62.5%)
Yes, I am attending a trade school, a technical institute or another non-university institution	1 (3%)	1 (6.3%)
Yes, I am attending university	9 (22%)	5 (31.3%)
Employment situation		
Full-time salaried employment	15 (38%)	5 (31.3%)
Part-time salaried employment	4 (10%)	1 (6.3%)
Self-employed	3 (8%)	1 (6.3%)
Receiving employment insurance benefits or COVID-19 emergency response benefit from government	5 (13%)	2 (12.5%)
Receiving social assistance or disability benefits	1 (3%)	1 (6.3%)
Sick leave	6 (15%)	3 (18.8%)
Maternity, paternity or parental leave	1 (3%)	1 (6.3%)
Unpaid work (e.g., childcare, volunteering, etc.)	2 (5%)	0 (0%)
Other	3 (8%)	2 (12.5%)
Working in the health and social services sector	13 (33%)	5 (31.3%)

* Values are *n* (%) unless otherwise indicated.

**Table 2 ijerph-20-03110-t002:** Depression, Anxiety and Stress Scale (DASS-21) scores of parents of children with disabilities from the province of Quebec (Canada) during the COVID-19 pandemic.

Scales	Sample Size *	Mean (SD)
Depression (/42)	33	13.5 (10.8)
Anxiety (/42)	34	10.5 (9.5)
Stress (/42)	36	19.2 (8.9)
Total scale (/126)	32	42.9 (27.0)

* Sample sizes correspond to parents without missing data on DASS-21 questions.

**Table 3 ijerph-20-03110-t003:** Answers of 16 parents of children with disabilities related to their parental role during the pandemic in Quebec province (Canada) ^1^.

Questions	Strongly Disagree	Disagree	Agree	Strongly Agree
The changes in the organization and the provision of services did not have an impact on the support I received in response to my requests.	4 (28.6%)	6 (42.9%)	4 (28.6%)	0 (0%)
I was able to establish an adequate educational support routine for my child.	3 (20.0%)	3 (20.0%)	6 (40.0%)	3 (20.0%)
The use of communication technologies (Facebook, Facetime, Skype, etc.) enabled my family to maintain relations with family and friends and other members of our circle.	1 (6.7%)	1 (6.7%)	5 (33.3%)	8 (53.3%)
Up to now, I’ve been able to respond to my child’s needs.	0 (0%)	1 (6.7%)	6 (40.0%)	8 (53.3%)
The suspension of activities among associations for the parents of children with disabilities or limitations has a big impact on many families needing support.	1 (6.7%)	2 (13.3%)	2 (13.3%)	10 (66.7%)
The suspension of weekend respite services has increased the burden of caring for my child for me and for other members of the family.	4 (26.7%)	2 (13.3%)	3 (20.0%)	4 (26.7%)
Health and social services workers have been able to respond to the needs of my child in spite of the changes in the organization of services provided.	3 (20.0%)	4 (26.7%)	5 (33.3%)	2 (13.3%)

^1^ Values are *n* (%). Sample sizes correspond to parents without missing data.

**Table 4 ijerph-20-03110-t004:** Answers of 40 parents of children with disabilities related to their level of solitude during the pandemic in Quebec (Canada) ^1^.

Questions	Never	Rarely	Sometimes	Often
I lack companionship	13 (37.1%)	9 (25.7%)	4 (11.4%)	8 (22.9%)
I feel left out	15 (42.9%)	6 (17.1%)	10 (28.6%)	4 (11.4%)
I feel isolated from others	10 (28.6%)	7 (20.0%)	11 (31.4%)	7 (20.0%)
I feel alone	12 (34.3%)	7 (20.0%)	6 (17.1%)	10 (28.6%)

^1^ Values are *n* (%). Sample sizes correspond to parents without missing data. Questions are four items selected from the UCLA Loneliness Scale (UCLA-LS).

**Table 5 ijerph-20-03110-t005:** Answers of 40 parents of children with disabilities related to their social life during the pandemic in Quebec (Canada) ^1^.

Questions	Strongly Disagree	Disagree	Agree	Strongly Agree
There are people I can depend on to help me if I really need it.	2 (5.7%)	4 (11.4%)	11 (31.4%)	18 (51.4%)
There are people who enjoy the same social activities I do.	2 (5.7%)	8 (22.9%)	15 (42.9%)	10 (28.6%)
I have close relationships that provide me with a sense of emotional security and well-being.	2 (5.7%)	4 (11.4%)	15 (42.9%)	14 (40.0%)
There is someone I could talk to about important decisions in my life.	0 (0%)	5 (14.3%)	12 (34.3%)	18 (51.4%)
I have relationships where my competence and skill are recognized.	1 (2.9%)	7 (20.0%)	12 (34.3%)	15 (42.9%)
There is a trustworthy person I could turn to for advice if I were having problems.	1 (2.9%)	3 (8.6%)	13 (37.1%)	18 (51.4%)
I feel part of a group of people who share my attitudes and beliefs.	4 (11.4%)	4 (11.4%)	14 (40.0%)	12 (34.3%)
I feel a strong emotional bond with at least one other person.	2 (5.7%)	2 (5.7%)	10 (28.6%)	20 (57.1%)
There are people who admire my talents and abilities.	2 (5.7%)	7 (20.0%)	12 (34.3%)	14 (40.0%)
There are people I can count on in an emergency.	1 (2.9%)	2 (5.7%)	9 (25.7%)	21 (60.0%)

^1^ Values are *n* (%). Sample sizes correspond to parents without missing data. Questions were from the Social Provisions Scale-10 items (SPS-10). The overall mean (SD) score was 17.60 (7.0) over 40.

## Data Availability

The MAVIPAN Database will be shared in accordance with the Canadian Institute of Health Research (CIHR) joint statement on sharing research data, the FAIR Guiding Principles for Scientific Data Management and Stewardship, and the norms established by the Ethic Committee of record. Data are available from the MAVIPAN study coordinator (contact via mavipan.ciussscn@ssss.gouv.qc.ca) for researchers who meet the criteria for access to confidential data.

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
