# Peer review of "Being a Parent of Children with Disabilities during the COVID-19 Pandemic: Multi-Method Study of Health, Social Life, and Occupational Situation"

_ijerph, 2023, doi:10.3390/ijerph20043110_

Round 1
Reviewer 1 Report
Dear authors:
The paper explores experiences of parents of children with disabilities during the COVID-19 pandemic in Quebec,
Acceptable: Introduction, methods, conclusions
Very good explanations in Tabs.
Very important results:
"Study reported that school closures had an impact on the physical and mental well-being. "
"Health professionals, policymakers and governments should
be mindful of the challenges of parents of children with disabilities highlighted in the present study."
It is important to provide correct references:
1. I can not read / open ....
2. Green, S. E. (2007) correct / SE incorrect/
....
10 compare : https://www.tandfonline.com/doi/abs/10.5172/mra.2013.7.1.2
check and correct
-----
We believe that these studies might be a relevant and and closely dependent on the current context. Comparative and complementary thoughts
:
Bokuniewicz, S. (2020). Tolerance of uncertainty and ambiguity of the situation and anxiety as a state and as a feature. Journal of Education Culture and Society, 11(2), 224–236. https://doi.org/10.15503/jecs2020.2.224.236
Tkacová H, Králik R, Tvrdoň M, Jenisová Z, Martin J.G. Credibility and Involvement of Social Media in Education—Recommendations for Mitigating the Negative Effects of the Pandemic among High School Students. International Journal of Environmental Research and Public Health. 2022; 19(5):2767. https://doi.org/10.3390/ijerph19052767
Murgaš, F., Petrovič, F., Maturkanič, P., & Kralik, R. (2022). Happiness or Quality of Life? Or Both?. Journal of Education Culture and Society, 13(1), 17–36. https://doi.org/10.15503/jecs2022.1.17.36
Maturkanič, P.; Tomanová Čergeťová, I.; Konečná, I.; Thurzo, V.; Akimjak, A.; Hlad, Ľ.; Zimny, J.; Roubalová, M.; Kurilenko, V.; Toman, M.; Petrikovič, J.; Petrikovičová, L. Well-Being in the Context of COVID-19 and Quality of Life in Czechia. Int. J. Environ. Res. Public Health 2022, 19, 7164. https://doi.org/10.3390/ijerph19127164
Reviewer 2 Report
Thank you for the opportunity to review "Experiences of parents of children with disabilities during the COVID-19 pandemic in the province of Quebec, Canada." This study explored the experiences of parents of children with disabilities during the pandemic. The paper was interesting and enjoyable to read. I commend the authors for their excellent layout and writing style. However, some revisions should be made before the paper is published in IJERPH. Please consider making the following revisions:
- List research questions. What exactly were you exploring/trying to answer with this study?
-Provide psychometric properties for measures.
- Did you pilot test the interview questions? If so, describe.
-Additional details are required for the participants and their children. What disabilities were the children diagnosed? Who are the five parents who participated in the interviews?
-Page 4. Italicize "n." Line 162 - Change to "The 16 parents who completed the additional questionnaire were..."
-Page 5. Lines 178-180. No need to describe scores in text if they are already in the Table. Just refer readers to Table 2.
-Throughout the results, frame the quotes by providing some context about the participant (i.e., was it a mother or father? what was their child's disability? etc.). It would be helpful to have context whether these quotes were from a mother of a 5 year old with ASD or a father of an 18 year old with a specific learning disability.
- Must include implications of this study's findings in the Discussion. What do these findings mean for parents of children with disabilities? What steps must be taken to help them?
With these changes in place, I believe the paper would be a good fit for IJERPH.
Reviewer 3 Report
Additional information is needed regarding the measures. First, they are described as validated, but very little information is given about them. The citations that are given refer to studies about the pandemic, some specifically about children with disabilities during the pandemic (more on ths in the next paragraph). Second, they are interpreted in relative terms, as if information is available about the scales beyond this study. Because there is no information beyond the study, the relative interpretation comes across as subjective, rather than as evidence.
The title of several citations indicates that research has been conducted about children with disabilities during the pandemic. This fact, combined with the very brief introduction, makes me wonder whether this study does fill a gap or simply reiterates previous research. Much of the information present is likely unique to this study - but not as unique as the claims made in the Conclusion. I also think the use of Bronfenbrenner's model was inappropriate. While parents are the microsystem of children's development, the focus of the paper was on adults not children. If this theory is applied, it should be introduced in the literature review. Additionally, I recommend that the use of Bronfenbrenner and Morris' most updated model is included. In this iteration of the theory, the chronosystem was introduced - and the COVID pandemic has been one of the most important events in our recent chronology of human development.
Reviewer 4 Report
Dear Authors,
I can perceive a high social relevance of your study. But before shifting to a further stage of the review, some questions concerning the design and research question must be clarified.
My main objective is with poorly formulated aim / research questions. Based on your aim / research question, I would say it is a qualitative study. However, your offered a mixed-design study with very important quantitative part (where qualitative part is rather explanatory for the quantitative one). This is inconsistent. Suggestions:
1. to publish it as a qualitative study - it should be clearly mentioned in the title / abstract (then, experiences is a proper descriptor for the title, as well as the aim/research question based on PICo components).
2. to publish it as a mixed-design study (as it is now), but then you should add separate research questions for quantitative and qualitative part and also indicate it clearly in the title / abstract. For quantitative part, there should be a question based on PIO components or its derivates (considering it is from a bigger observational study, as you mention in the text). I would consider also adding primary outcomes from quantitative part into the study title.
If you want to publish it as mixed design study, I also consider what is actually the main focus - data from objective outcome measures? (and then thematic analysis from interviews is used to explain quantitative findings) or should it be on the same level of relevance? (and then you should integrate quantitative and qualitative findings in an appropriate way)
Some other concerns:
Poorly defined eligibility criteria (Methods section): use the main elements of your research questions (Participants, Phenomena of Interest, Context, Outcomes...) to clearly explain what was your original plan for the study.
What are the disabilities and level of severity of the participants? you mention in the limitation, there were several types of disability. Because there are significant differences in the impact (according to the type / severity of disability) we should at least know what was the diagnosis of the child.
Then, is it possible what questionairres were completed during the lockdowns? How could the development of pandemic situation and release of restrictions inlfuence the data?
It would be better to include information about MAVIPAN study into the Introduction section, and explain there how and why you decided to carry out also this smaller study.
Round 2
Reviewer 1 Report
accepted